# Lipophilic Toxins in Chile: History, Producers and Impacts

**DOI:** 10.3390/md20020122

**Published:** 2022-02-04

**Authors:** Patricio A. Díaz, Gonzalo Álvarez, Gemita Pizarro, Juan Blanco, Beatriz Reguera

**Affiliations:** 1Centro i~mar (CeBiB), Universidad de Los Lagos, Casilla 557, Puerto Montt 5480000, Chile; patricio.diaz@ulagos.cl; 2Departamento de Acuicultura, Facultad de Ciencias del Mar, Universidad Católica del Norte, Larrondo 1281, Coquimbo 1781421, Chile; gmalvarez@ucn.cl; 3Centro de Investigación y Desarrollo Tecnológico en Algas (CIDTA), Facultad de Ciencias del Mar, Universidad Católica del Norte, Larrondo 1281, Coquimbo 17811421, Chile; 4Centro de Estudios de Algas Nocivas (CREAN), Instituto de Fomento Pesquero (IFOP), Enrique Abello 0552, Punta Arenas 6200000, Chile; gemita.pizarro@ifop.cl; 5Centro de Investigacións Mariñas (Xunta de Galicia), Apto. 13, 36620 Vilanova de Arousa, Pontevedra, Spain; juan.carlos.blanco.perez@xunta.gal; 6Centro Oceanográfico de Vigo (IEO, CSIC), Subida a Radio Faro 50, 36390 Vigo, Pontevedra, Spain

**Keywords:** *Dinophysis acuminata*, *Dinophysis acuta*, lipophilic toxins, diarrhetic shellfish poisoning, harmful algal blooms, HAB monitoring, Chile

## Abstract

A variety of microalgal species produce lipophilic toxins (LT) that are accumulated by filter-feeding bivalves. Their negative impacts on human health and shellfish exploitation are determined by toxic potential of the local strains and toxin biotransformations by exploited bivalve species. Chile has become, in a decade, the world’s major exporter of mussels (*Mytilus chilensis*) and scallops (*Argopecten purpuratus*) and has implemented toxin testing according to importing countries’ demands. Species of the *Dinophysis acuminata* complex and *Protoceratium reticulatum* are the most widespread and abundant LT producers in Chile. Dominant *D. acuminata* strains, notwithstanding, unlike most strains in Europe rich in okadaic acid (OA), produce only pectenotoxins, with no impact on human health. *Dinophysis acuta*, suspected to be the main cause of diarrhetic shellfish poisoning outbreaks, is found in the two southernmost regions of Chile, and has apparently shifted poleward. Mouse bioassay (MBA) is the official method to control shellfish safety for the national market. Positive results from mouse tests to mixtures of toxins and other compounds only toxic by intraperitoneal injection, including already deregulated toxins (PTXs), force unnecessary harvesting bans, and hinder progress in the identification of emerging toxins. Here, 50 years of LST events in Chile, and current knowledge of their sources, accumulation and effects, are reviewed. Improvements of monitoring practices are suggested, and strategies to face new challenges and answer the main questions are proposed.

## 1. Introduction

Among the different kinds of Harmful Algal Blooms, those of toxin producing microalgae pose the main threat to shellfish exploitation worldwide. These microalgae are taken up by filter feeders and transferred to higher trophic levels through the food web. The most common toxin groups accumulated by shellfish in temperate seas are those causing paralytic shellfish poisoning (PSP), amnesic shellfish poisoning (ASP) and lipophilic shellfish toxin (LST) syndromes. The latter (LST) includes the old DSP complex, i.e., diarrhetic shellfish poisoning (DSP) toxins, pectenotoxins (PTXs) and yessotoxins (YTXs) and the azaspiracids (AZAs). All these toxins, produced by different dinoflagellate genera, are under regulation and are co-extracted with the organic solvents used with standard methods for LT determination.

The most important primary source of LST worldwide are several species of the genus *Dinophysis*, which can produce one or two kinds of these toxins: (i) okadaic acid (OA) and dinophysistoxins (DTXs)—i.e., the okadaic acid group or okadaates (OAs), the only diarrheogenic toxins in the old DSP complex—(Figure 1) and (ii) PTXs, the toxicity of which has long been a matter of controversy (Figure 2) [1,2]. *Phalacroma rotundatum* and *Ph. mitra* have been confirmed to contain OAs and PTXs, but the production of toxins de novo by these species has not been proved in cultures. Furthermore, there is field evidence which suggests they do not produce toxins but obtain them from their prey [3].

Yessotoxins (YTXs) (Figure 3) have been confirmed to be produced by *Protoceratium reticulatum* [4], *Lingulodinium polyedra* [5] and two species of *Gonyaulax*, *G. spinifera* [6] and *G. taylorii* [7]; and AZAs (Figure 4) by species of the genera *Azadinium* and *Amphidoma* [8].

In addition, there are other emerging lipophilic toxins not subject to regulations, the “fast acting toxins” (FAT) [10]). These include the spirolides (SPXs), the gymnodimines (GYMs) and the pinnatoxins (PnTXs), produced by *Alexandrium ostenfeldii*, *Karenia* species and *Vulcanodinium rugosum*, respectively (Figure 4).

The toxicity of the causative organisms depends on their toxin profiles and content, which are clearly species-dependent and also vary intra-specifically. Toxin profiles and cellular content of *Dinophysis* species have shown large intraspecific variability [11], but little is known about this aspect of PTXs and AZAs producers. These differences in toxic potential cause species’ strain specific impacts on shellfish contamination and on the duration of toxic events.

The impacts of toxic microalgae blooms are particularly severe in coastal areas with intensive cultivation of mytilids, such as the Galician Rías [12], Northwest Spain and the Region of Los Lagos in Northern Chilean Patagonia [13]. Mediterranean mussel (*Mytilus galloprovincialis*) cultivation in the former has reached a stable production of around 250,000 t per year, and according to the last report from the Spanish Association of Aquaculture Producers, inter-annual differences in mussel production in the region have been

Related to the greater or lesser incidence of toxic microalgal events that disturb regular shellfish harvesting [15]. In contrast the Chilean mussel *Mytilus chilensis* (“choritos”) production has been doubled in the last ten years in Chilean Patagonia, reaching an annual yield of 400,000 t. This makes Chile the second world producer of mussels after China. In addition, about 25,000 t of other bivalve species of high market value are collected from natural shellfish beds. Among these, the most valuable resource comprises the scallop *Argopecten purpuratus* (“ostión del norte”) in the region of Coquimbo (Table 1).

DSP intoxications are underreported worldwide because (i) DSP is a non-fatal syndrome and symptoms are confused with bacterial gastroenteritis; (ii) monitoring is usually implemented when new aquaculture facilities are established; (iii) DSP events are overlooked unless there are intoxicated consumers, and/or monitoring of this group of toxins is disrupted when long quarantines are enforced due to bivalve contamination with high levels of more lethal (e.g., PSP) toxins [11].

Chile has not been an exception. Early reports from the 1970’s, most of them in the grey literature, described major acute gastrointestinal outbreaks affecting hundreds of consumers before any monitoring system was established. Given the magnitude of the Chilean coast, extending from 17° S to 56° S latitude (Figure 5), and the complexity of the Patagonian coastline, with multiple fjords, islets and inland seas with difficult access, make high spatial-temporal-resolution sampling an impractical task. Consequently, greater efforts have been dedicated to the control of the most lethal toxins, i.e., those causing PSP [16]. Observations of lipophilic toxins have been more sporadic, making it more difficult to evaluate their fate through food webs and their actual impacts. Emerging toxins have been added up to the total LST fraction during severe outbreaks in southern Chile. The mass mortality of well-boat salmon and benthic resources during summer 2017–2018 in Magallanes and Aysén regions, associated with a bloom of *Karenia* species, drew attention to the threat these toxins may pose, not only to public health and commercial resources but also to marine ecosystems [17,18,19,20].

This work reviews historic antecedents of LST outbreaks in Chile, new events related to emerging toxins, and their relation to potential producers, and points to the apparent poleward shift in the distribution of DSP toxins on the basis of the distribution of acute outbreaks in the 1970’s and 1980’s and the current situation. Finally, characteristics of LST accumulation in the main commercial species and their impacts are discussed in the light of recent regulatory changes in the European Union [21], the main destination for Chilean exports.

## 2. Historical Overview

Since the early 1960’s, oceanographic cruises (MARCHILE series and others) have been carried out, in particular off the upwelling influenced regions in northern Chile and in Chilean Patagonia [22,23,24]. Most of these cruises were not planned for the detection of toxin producing microalgal species. Nevertheless, phytoplanktologists at the Chilean marine research centres in those days provided comprehensive information on microphytoplankton species distributions, including those related to shellfish poisoning outbreaks in the 1970’s and 1980’s. A compilation of water discolorations or red tides (high biomass blooms and potentially toxic species), including the historic description of a red tide of the ciliate *Mesodinium* by Charles Darwin, were presented in a UNESCO workshop in 1980 about Eastern Pacific red tides [25]. Red tides of ciliates (*Mesodinium rubrum*) associated with upwelling fronts and high biomass blooms of dinoflagellate species of the genera *Prorocentrum*, *Scrippsiella* and *Ceratium* were predominant in the north, from (20–32° S) and those responsible for toxic outbreaks (*Alexandrium catenella*, *D. acuta*) in the Patagonian region (42–55° S) [22,23,24,26]. Recurrent severe PSP outbreaks in Patagonia triggered target-species oriented cruises in the 1990’s [27,28,29].

Chile has a long history of DSP events due to lipophilic toxins affecting consumers in the three southernmost regions of the country (Los Lagos, Aysén and Magallanes). The first gastrointestinal outbreak in Southern Chile, and the first in the world to cast suspicion on *Dinophysis* as the toxic agent, dates back to 1970. That year, over 100 people were severely affected by gastrointestinal disorders after eating ribbed mussels or “cholgas” (*Aulacomya atra*) from Reloncaví Sound (Los Lagos Region). These mussels were presumably contaminated with phycotoxins later associated with an intense bloom of *D. acuta* [30,31]. Guzmán and Campodonico [30] quoted a personal communication from J. Hermosilla, confirming that in 1970 and 1971, red tides (discolorations) formed by *Dinophysis* were observed off Puerto Montt (41°30′ S), Los Lagos Region; these red tides were associated with gastrointestinal disorders among local consumers of various shellfish species. New intoxications in the same region occurred in 1979 and 1986. Lembeye et al. [32] described the acute response in intoxicated consumers of *Au. atra* and *My. chilensis* and plankton observations between March and July 1979. *D. acuta* was the overwhelmingly dominant species in the microplankton, and shellfish exposed to the bloom had their digestive track full of empty theca of that species. Subsequent reports, in 1989 and 1994, were again associated with *D. acuta*, but in Jacaf Fjord, Aysén Region, 240 km south of Reloncaví Sound. Nevertheless, the first major DSP outbreak in Aysén, which affected 520 consumers, was in 1991 in the Chonos Archipelago. This outbreak increased the authorities’ awareness concerning the impact of lipophilic toxins on public health, and the first steps were taken to establish a monitoring programme. The first chemical analyses were carried out in collaboration with Japanese experts and revealed an important contribution of okadaates (mainly DTX1 followed by OA) in contaminated shellfish [33].

Campodonico and Guzmán [34] described in January 1973 what may have been the first azaspiracid poisoning (AZP) event. The authors mentioned concentrations of *Amphidoma* sp. up to 678,000 cells L^−1^ off Punta Arenas, Magallanes region, during unusually warm sea surface temperature (SST) and stratified water-column conditions. Symptoms in rats used for the bioassays were different from those caused by PSP toxicity and shellfish had the digestive tract full of empty *Amphidoma* theca. This bloom developed after a severe PSP outbreak. Therefore, interferences in the bioassays with remains of PSP toxins could not be discarded.

Yessotoxins were first detected by Yasumoto and Takizawa [35] in mussels from the Chonos Archipelago. Further detection of YTXs in different parts of the country and identification of the producing microalgae in Chile had to wait until chromatography coupled to mass spectrometry techniques were implemented. *Protoceratium reticulatum* blooms were related to YTXs in shellfish [36] followed by *Gonyaulax spinifera, G. polygramma* and the addition of a new *Gonyaulax* (*G. taylorii*) to the list [7].

## 3. Monitoring of Lipophilic Shellfish Toxins in Chile

The harmful algae monitoring system in Chile is diverse and adapted to different socio-economic demands along its extensive (over 4000 km) coast. Currently, three different toxin monitoring programmes are operational in Chile. Two of these, the first run by the Health Ministry and the second by the Fisheries Development Institute (IFOP), are focused on the protection of public health and rely on mouse bioassay (MBA) for toxin LST detection and quantification [37]. The third one, the National Shellfish Sanitation Programme (PSMB), run by SERNAPESCA (National Fisheries and Aquaculture Service), is focused on control of the safety of shellfish exports. Its application is optional and adapted to the requirements of importing countries [38]. Many of these are European countries subject to common European Commission regulations [21,39].

The *Health Ministry Monitoring Programme*, established to safeguard public health of local populations, coordinates activities at regional health centres. This national-scale programme includes monthly LST detection in shellfish by a modified version of the Yasumoto MBA from 1984 [40,41]. The results are used to guide enforcement of sanitary closures when the permissible limits for human consumption of any type of marine biotoxin are exceeded. Since 2012, intensification of LST events in Aysén and Magallanes has become a serious problem. This new situation has led to increased frequency of mouse bioassay testing by regional health authorities (e.g., SEREMI Salud Magallanes) during unusually severe outbreaks [42,43].

The *Harmful Algal Bloom Management and Monitoring Programme* has been carried out by the Fisheries Development Institute (IFOP, Instituto de Fomento Pesquero) since 2006 in the three southernmost regions of Chile. The main objective is the detection and quantification of shellfish toxins posing risks to human health and the identification and quantification of potentially harmful microalgal species, the primary sources of marine biotoxins that can cause shellfish toxicity or ecosystem disruption. This programme includes 146 fixed monitoring stations (43 in Los Lagos; 57 in Aysén; 46 in Magallanes), which are sampled monthly and/or weekly [16]. Sampling frequency can be increased upon request from the Fisheries and Health authorities. Prompted by the first PSP event on the oceanic coast of Los Lagos and Los Ríos regions in summer 2016 [44,45], a new sampling grid was approved in 2018 including oceanic coast stations between Bío Bío and Los Lagos [46].

The *National Shellfish Sanitation Programme* (PSMB, Programa de Sanidad de Moluscos Bivalvos) is coordinated by SERNAPESCA. This programme, implemented in 1989 in Northern Chile and in 1996 in the main mussel aquaculture site in Chiloé, Los Lagos, includes sampling for toxins determination in mussels from over 100 stations. In 2015, the MBA protocol for determination of LST was replaced with the EU harmonized liquid chromatography coupled to the mass spectrometry (LC-MS) method [39]. Sampling frequency is increased during periods when toxin levels exceed pre-established concentrations and/or after detection of toxic phytoplankton species. The PSMB also includes concurrent phytoplankton monitoring that complements toxicity determinations and provides an early warning to stakeholders [47]. Unlike the two previous programmes, which were financed by the Chilean government, the PSMB programme is financed by private companies that wish to export their products to the European Union and other countries.

It is well known that in the mouse bioassay for lipophilic toxins, total toxicity is quantified, and results are given in OA equivalents. Specification of the contribution of each toxin group to total toxicity in the extract is not possible with this method. In addition, the MBA is very unspecific, requires a long observation time of the mice, and may give false positives in samples with high fatty acid content. Different symptoms in the mice may be distinguished by experienced personnel [48]. However, during exceptional multi-specific blooms, which require the analysis of very complex extracts, complementary analyses by chromatography-mass spectrometry (LC-MS) have been essential. That was the case in Magallanes, during summer 2012–2013, when high concentrations of microalgae producers of DSP, PTXs, YTXs and spirolides co-occurred [42].

## 4. Lipophilic Shellfish Toxin Producers in Chile: Distribution and Toxic Events

Information about toxin profiles and content in Chilean strains of different species of *Dinophysis* and *Gonyaulax*, *Lingulodinium polyedrum* and other LST producers, and their contribution to the toxic outbreaks in Southern Chile is largely unknown.

Mouse bioassay results often correspond to multi-specific blooms of *Dinophysis* with *P. reticulatum* and temporal resolution of the phytoplankton sampling is clearly inadequate to establish relationships between cellular concentrations and toxins in shellfish. Most of the toxinological information below is based on events where one single species was overwhelmingly dominant in the microplankton fraction. Some information has been obtained from LC-MS analyses carried out by private companies (export controls) or by national and international research and cooperation projects. In addition, impacts distinct from shellfish contamination, such as mass mortalities potentially linked to LST producers and or to emerging toxins, some of them of unknown origin, are included.

### 4.1. Dinophysis and Its Lipophilic Toxins

Potentially toxic species of *Dinophysis* are well represented in Chilean coastal and oceanic waters (see Table 2 and references therein). Some species, e.g., *D. caudata* and *D. tripos*, show a positive latitudinal gradient from south to north. Others, e.g., *D. norvegica* and *D. truncata*, are restricted to cold Austral waters; *D. acuta* occurs mainly in the three southernmost regions (Los Lagos, Aysén and Magallanes) and different morphotypes of the *D. acuminata* complex are found along the entire Chilean coast (Figure 6).

#### 4.1.1. The *Dinophysis acuminata* Complex

This term is used to designate morphologically close *Dinophysis* species with a high intraspecific variability in size and shape, but almost identical partial sequences of their small sub-ribosomal unit plus the internal transcribed spaces (ITS1-5.8S rRNA-ITS2) and mitochondrial (*cox1*, *cob*) genes [87,88]. Species included in this complex in Europe and Eastern USA are *D. acuminata*, *D. ovum* and *D. sacculus*. The last two species were listed in earlier studies on phytoplankton distribution in Northern Chile [23,24,69] but unfortunately illustrations and cellular concentrations of the species were not included. The Chilean morphotypes in the last two decades have been labelled as *D. acuminata* complex, *D.* cf. *acuminata* and *D.* cf. *ovum* but as in other parts of the world, there is uncertainty in the classification of some morphotypes that sometimes include misclassification of intermediate or small forms of the polymorphic life cycle of a different species of *Dinophysis*. Figure 7 illustrates the variety of specimens labelled as components of the *D. acuminata* complex species from different Chilean regions. Some of these images with almost triangular antapical silhouettes, may be intermediate cells of *D. norvegica*. It is probably not coincidental that these morphotypes are all from the southernmost region of Magallanes.

*Dinophysis acuminata* is ubiquitous on the Chilean coast and the most frequently reported species of the genus at scallop *Ar. purpuratus* aquaculture sites in Coquimbo (~30° S) (Eduardo Uribe pers. comm.). Harvesting closures due to DSP were not enforced in that region until October 2005, when scallop exploitation was affected by lengthy preventive closures due to the detection of *D. acuminata* and, in a few cases, positive results of the DSP bioassays (reviewed by [13]). Blanco et al. [54] showed that during that event, *D. acuminata* was present most of the time along 350 km of the coast, with cell maxima of 600–900 cells L^−1^ in different bays. Samples for quantitative analyses were taken with a tube sampler from 0 to 15 m. This species frequently aggregates in narrow layers close to the surface [52,53,89] and differences between 1 and 2 orders of magnitude are easily observed between tube and bottle sample counts of *Dinophysis* [90]. It is thus highly probable that very high concentrations of *D. acuminata* occurred during that event. Analyses, by high performance liquid chromatography (HPLC) coupled to mass spectrometry (MS) of picked cells, plankton concentrates, and shellfish exposed to the bloom, were consistent with a toxin profile containing only PTX2 and undetectable levels of other toxins. The analysed cells contained extremely high levels of PTX2 (toxin quota of 180 pg cell^−1^) and probably trace levels of PTX1 or PTX11. The toxin profiles found in molluscs from the area (*Ar. purpuratus* and *Mesodesma donacium*) were, as expected from the plankton profile and the biotransformation characteristics of the affected bivalve species, PTX2 and PTX2sa. These results agree with observations by Krock et al. [55] two years later during a survey in Arica (~18° S) in summer 2007–2008. Plankton hauls (20–70 µm size fraction) from that survey analysed by LC-MS revealed the occurrence of PTX2, PTX11, and PTX2sa with maximal concentrations of 7604, 134, and 39 pg L^−1^, respectively. The survey coincided with the occurrence of *D. acuminata* (up to 2400 cells L^−1^) and *Ph. rotundatum*. In a later study during 2009–2010, *D. acuminata* was frequently present (75% of weekly samples) in Coquimbo Bay where it reached a maximum of 2100 cell L^−1^ [49] (Figure 7A). Analyses of *Me. donacium* samples showed only PTX2, its seco-acid (PTX2sa) and PTX2sa esters.

Dense populations of *D. acuminata* have also been found in Arauco Gulf, Bío Bío, at the southernmost limit of the Humboldt Current upwelling system. Benthic resources in this region are the second most important in the country after Los Lagos in terms of gross catches and market value (Table 1). An exceptional bloom of *D. acuminata* reached 38,400 cells L^−1^ in (0–10 m) water column integrated tube samples (Figure 7B) and was associated with the first report of PTX-2 in hard razor clams *Tagelus dombeii* [51]. The toxin analysis of *T. dombeii* extracts showed only PTXs.

In the Los Lagos region, *D. acuminata* is present all year round and has a long growing season, from spring to autumn. High-density blooms (>10^4^ cells L^−1^) of this species have been found in Reloncaví Fjord, usually in October to March, forming thin layers above or within the pycnocline [52,57]. Toxin analyses of plankton concentrates during blooms of *D. acuminata*, of single cell isolates, and recently of laboratory cultures, suggest the existence of strains with different profiles: some strains produce only PTXs, and others include DTX1 and PTXs. In 2005–2006 (December to February), passive samplers (SPATT resins) were suspended at 10 stations along two transects on the east coast of Chiloé and one across the mouth of Reloncaví Fjord. PTX2 and derivatives were the only toxins found in the Chiloé transects, but DTX1 (with co-occurring PTX2) predominated in the Calbuco Island and other stations at the mouth of Reloncaví [79]. *Dinophysis acuminata*, abundant in the region at this time of the year, was the most likely producer of these toxins (Figure 7C).

Alves de Souza et al. [52] followed changes in *Dinophysis* populations and associated toxins in plankton concentrates and in shellfish during two growing seasons (late spring to early summer 2007–2008 and early spring to late summer 2008–2009). Chromatographic analysis with fluorescent detector (HPLC-FD) of the mussels’ digestive glands (HP) in early summer (December) 2007 revealed DTX1 and DTX3 (280 and 239 ng g^−1^ HP, respectively) in mussels associated with low numbers (<500 cells L^−1^) of D. acuminata. These *Dinophysis* cells possibly corresponded to residual populations from a preceding bloom, and one month later only DTX3 (184 ng g^−1^ HP), a mollusc biotransformation derivative, was detected. Results from the IFOP monitoring confirmed a high relative abundance of *D. acuminata* and the dominance of dinoflagellates, the preceding spring, in the area. DTXs dropped by February and remained low, even when an exceptionally dense bloom of *D. acuminata* developed in late March 2008. Those analyses by HPLC-FD did not search for PTXs. During the next season, LC-MS analyses showed very moderate levels of PTX2 in plankton concentrates and shellfish, coinciding with increasing numbers of *D. acuminata*. There was always a minor proportion of other *Dinophysis* species (e.g., *D. subcircularis*) and blooms between consecutive sampling may have been missed. In addition, mussels always had trace levels of DTX1 and DTX3.

In a parallel study, cells of *D. acuminata* were isolated from samples collected in Reloncaví (coinciding with the March 2008 bloom in the second part of the previous study) and established in culture. Analyses (10^5^ cells) of culture samples revealed only PTX2 and no signs of okadaates [58].

Between October 2015 and September 2018, the PSMB monitoring programme of toxins in mussels for export included LC-MS analyses of over 11,000 samples collected from 67 stations of the monitoring grid in Chiloé Inland Sea [47]. Analyses, according to the EU protocols [39], revealed that 965 samples (about 8.8%) were found to contain LST above the level of detection (LOD) but PTX2 was detected in only 17% (174 samples), i.e., 1.5% of the total, and traces of DTX1 and AZA in 14 samples. YTXs, detected in 773 (>80%) of the 965 samples with LST, and associated with the occurrence of *P. reticulatum*, were the dominant lipophilic toxins. In all cases, concentrations of *Dinophysis*-related toxins were below 160 µg (OA + DTXs + PTXs) per kg and those of YTXs below 3.75 mg YTX+45 OH-YTX+45 OH-homoYTX) per kg of shellfish meat, i.e., the EU regulatory levels (R.L.) for these two LST groups. The presence of PTX2 was related to *D.* cf *acuminata* but suggestions about the potential source of DTXs and AZA were not given.

In Aysén, *D. acuminata* is endemic and blooms of variable intensity and duration occur every year between spring and autumn. *D. acuminata* may co-occur with *D. acuta*, but their niches are vertically segregated [53]. In samples with an overwhelming dominance of *D. acuminata* (Figure 7D) in Puyuhuapi Fjord, LC-MS analyses revealed a cellular content of PTX2 alone.

In Magallanes, *D. acuminata* is the most frequent and abundant *Dinophysis* species. Detection of DSP toxins in shellfish and symptoms in local consumers from that region were not reported until 1998 [59]. Targeted sampling followed after observations of a bloom of *D. acuminata* in Estero Núñez (53°19′ S). Analyses by HPLC revealed the presence of DTX, but not of OA in shellfish; PTX2 was not included in the analyses. The ciliate *M. rubrum* (now recognized as prey of *Dinophysis*), was the most abundant organism in samples with dominance of dinoflagellates. The authors drew attention to the occurrence of two clearly distinct morphotypes of *D. acuminata* and illustrated them. More recently, during an intense multi-specific bloom in 2012, picked cell analysis of *D. acuminata* corresponding to the morphotype shown in Figure 7E,F showed the presence of DTX1 and PTX2; the same toxins in addition to PTX2sa were found in mussels [42].

It is thus clear that *D. acuminata* produces pectenotoxins along the entire Chilean coast. A production of toxins of the OA group in addition to PTXs in strains from Patagonia was suspected during the 2012 event from the association of *D. acuminata* blooms and the DTX1 and PTX2 profiles of exposed shellfish. This was later confirmed in picked cell analysis [42,43]. D acuminata strains with only PTX2 are found on northern and central Chilean coasts, within the limit of the Humboldt Current upwelling system [51,54]; a mixture of the two strains can be found in Los Lagos and Aysén (northern Patagonia), although segregated in time and with the strains producing OA less and less frequently observed in Los Lagos in recent years. Strains with one (PTXs) or two (OA + PTXs) groups of toxins seem to be common in Magallanes. The occurrence of strains producing only PTXs throughout Patagonia has recently been demonstrated in cultures established from specimens isolated from Los Lagos, Aysén and Magallanes [60].

The existence of different strains of *D. acuminata*, with profiles containing one or two groups of lipophilic toxins, have been reported in Perú, within the same (Humboldt Current) upwelling system as northern Chile [91]. The Peruvian strains had low content of OA (<1 pg cell^−1^) and/or only PTX2 (1–11 pg cell^−1^); two distinct morphotypes were illustrated; picked cell samples analysed contained a mixture of the two. Strains with the two kinds of profiles are also common in Denmark [92,93] and Sweden [94], in Skagerrak-Kattegat and northern North Sea waters.

#### 4.1.2. *Dinophysis acuta*

*Dinophysis acuta* is the second most abundant *Dinophysis* species in Chile but has a narrow growing season (late summer to autumn) and latitudinal distribution (Patagonia) compared with *D. acuminata*. However, *D. acuta* is by far the most harmful *Dinophysis* species in Southern Chile. Blooms of *D. acuta* are always associated with detection of OAs and PTXs in shellfish. Dense populations of this species have been described associated with density fronts in the mouth of Aysén Fjord [65].

Chilean strains of *D. acuta* show stable morphology in all areas where it has been reported. Its large hypothecal plates have a more angular contour and narrower dorso-ventral depth than western European morphotypes (Figure 8A). Despite its abundance and frequency in Southern Chile, nothing is known about its genetics, except some recent information on (klepto)plastid sequences (23S rDNA) from specimens isolated from Puyuhuapi Fjord, Aysén [64].

High resolution observations in recent years have shown record concentrations (up to 6.5 × 10^5^ cells L^−1^) of *D. acuta* in thin layers in the pycnocline, in particular in years with climate anomalies leading to intensified thermohaline stratification in summer (January to March). Puyuhuapi Fjord has been identified as the epicentre of regional events (Figure 9). Quasi monoalgal blooms there allowed toxin analyses of field populations where *D. acuta* represented 99% of the total *Dinophysis* population [62]. The profile consisted of OA (10–100 pg·cell^−1^) and DTX1 (2–20 pg·cell^−1^) in addition to PTX2 (4–90 pg·cell^−1^). These, in addition to PTXsa derivatives, were the main toxins accumulated in mussels. Current knowledge thus indicates that the primary source of OA on the Chilean coast is *D. acuta*.

There is growing evidence of a poleward shift of *D. acuta* populations in Chilean Patagonia. *D. acuta* is becoming almost a rare species in Los Lagos, the site of outbreaks in the 1970’s and 1980’s [61]; this species no longer poses a risk to the mussel industry there. In contrast, there is an apparent intensification of *D. acuta* events in Magallanes [63].

#### 4.1.3. Other *Dinophysis* spp.

There are other potentially toxic species of *Dinophysis* and *Phalacroma* (Figure 8I) in Chilean waters, but no information about their toxin profiles and contents, nor about their contributions to toxic outbreaks (Table 2).

*Dinophysis caudata* (Figure 8E) and *D. tripos* (Figure 8B) have been reported in all floristic lists of phytoplankton species derived from oceanographic surveys in Northern Chile [95]. They have also been cited as co-occurring species during blooms of *D. acuminata* in Coquimbo. *D. caudata* and *D. tripos* are the main primary producers of LST (OAs and PTXs) in tropical and subtropical waters, e.g., in The Philippines, [96]. The two species often present strains containing only PTXs, but occurrence in the same location of strains containing OA and DTXs in addition to PTXs has been reported. That is the case of *D. caudata* in Galician waters [97]. Conversely, LC-MS analyses of picked *D. caudata* cells from two different bays, Sechura and Samanco in Northern Perú during different years (2017 and 2018) contained only PTX2 [91]. These authors illustrated morphotypes of *D. caudata* f *abbreviata*, the most frequent form observed in warm-temperate waters [98], which coincides with the Chilean morphotypes from tropical northern latitudes but with cold upwelling waters in Coquimbo (Figure 8E).

*Dinophysis caudata* (f *abbreviata*) and *D. tripos* are frequently cited as accompanying species during blooms of *D. acuta* and *D. acuminata* in Northern Patagonia. It is assumed that these species do not contribute significantly to shellfish toxicity in the region.

*Dinophysis norvegica*, abundant in fjords and embayments in Northern Europe, Northeast USA and Canada, is probably under-reported in Southern Chile. Some micrographs from Southern Chile, including some illustrations in Uribe et al. [59], drew attention to different morphotypes of *D. acuminata*; these may be misidentified intermediate forms of *D. norvegica* (Figure 8C). This coastal cold-water species is morphologically very variable and a source of taxonomic uncertainty in monitoring programmes where blooms of *D. acuminata* and *D. norvegica* co-occur.

*Dinophysis subcircularis* (Figure 8D), was cited by Balech [99] in coastal waters of Argentina (42–48° S) as a possible synonym of *D. punctata*. Moderate levels (<500 cells L^−1^) of both species were reported from late spring to summer 2007–2008 in Reloncaví Fjord [52]. Neither species has been tested, anywhere in the world, for possible lipophilic toxin content.

*Phalacroma rotundatum* (= *D. rotundata*; Figure 8I) is a cosmopolitan heterotroph, present along the entire Chilean coast. Its capacity to produce toxins de novo has not been demonstrated. Field populations may be toxin-free, or they may contain toxins that reflect the profiles of accompanying *Dinophysis* species. Their toxin content may be the result of secondary herbivory, i.e., Phalacroma feeding upon ciliates that have previously eaten toxic *Dinophysis* [3]. They can therefore act as vectors, transferring toxins to shellfish and other filter-feeders. However, there is no evidence of a bloom of *Ph. rotundatum* alone (with no accompanying *Dinophysis*) related to shellfish intoxication.

Other species recorded during the last 20 years of monitoring and research in Chile are *D. fortii* (North to Central Chile; Figure 8F) and *D. truncata* (Magallanes; Figure 8G), but information about their distribution and abundance or about their toxin profile is not provided.

### 4.2. Protoceratium reticulatum and Other Yessotoxin Producers

Yessotoxin producers in Chile include *P. reticulatum*, *G. polygramma* and *G. spinifera*. Their contribution to positive results in MBA tests is difficult to estimate because they usually co-occur with *Dinophysis* species.

#### 4.2.1. *Protoceratium reticulatum*

*P. reticulatum* (*Gonyaulax grindleyi*), a bloom forming toxin producer, is widely distributed on the Chilean coast (Figure 10). High densities have been reported from Chipana (Arica) to Magallanes. Unlike *Dinophysis* species, *P. reticulatum* has a benthic resting stage, enabling it to alternate between pelagic and benthic habitats in response to environmental conditions. Viable cyst beds in Chiloé Inland Sea and in coastal and oceanic waters in Los Lagos and Aysén are known [71,72,73,74,75].

In the north, an intense bloom in summer 2007 [36] was initiated by shelf populations advected to the coast during upwelling relaxation. The bloom spread from Chipana (Arica), with a cell maximum of 100,000 cells L^−1^ [76], to Bahía Mejillones (Antofagasta), 180 km south, where maximum numbers (383,520 cells L^−1^) were reported by Rossi and Fiorillo [77]. *P. reticulatum* cell contents of YTXs ranged between 0.2 and 0.4 pg cell^−1^. During the bloom, there was a high mortality of scallops (*Ar. purpuratus*) in an aquaculture farm in Bahía Mejillones. The highest mortality was detected in seed scallops (12 ± 2 mm shell height) which were suspended at 10 to 15 m depth. The mortality was attributed to hypoxic and anoxic conditions caused by decay of an earlier bloom, but *P. reticulatum* and its YTXs may have contributed to the mortality, as demonstrated in other studies [100,101]. These results agree with observations in December the same year by Krock et al. [55] and the presence of YTX in plankton hauls off Bahía Arica, with a maximum YTX concentration of 422 pg L^−1^.

*Protoceratium reticulatum* is a significant threat to the mussel industry in Los Lagos (Figure 10). In summer 2009, there was a positive correlation between a moderate *P. reticulatum* bloom (2.2 × 10^3^ cell L^−1^) and moderate to high concentrations of YTX in shellfish (51–496 mg kg^−1^) and in plankton concentrates (3.2 ng L^−1^) in Reloncaví [52]. In summer 2015, densities of *P. reticulatum* close to 12 × 10^3^ cell L^−1^ were reported from Bahía Huelmo—an important mussel cultivation area within Reloncaví—associated with YTX concentrations above the EU R.L. (3.75 mg YTX eq. kg^−1^), and harvesting closures were enforced (Miriam Seguel, pers. commun.). Results from 3 years of monitoring in Los Lagos, including the most valuable mussel aquaculture site in Chiloé [47], identified *P. reticulatum* as the most frequent cause of harvesting bans. LST were detected in 8.7% (n = 965) of all samples (n = 11,100) and yessotoxin, associated with *P. reticulatum*, accounted for 80.1% of them (773/965). These authors found a north to south gradient in bloom intensity and YTX levels, with maximum numbers near Calbuco islands at the mouth of Reloncaví.

The presence of YTX was associated with moderate numbers of *P. reticulatum* (max. 1700 cells L^−1^) in summer 2018 in Puyuhuapi Fjord, Aysén [78].

In spring 2010, *P. reticulatum* was associated with YTXs in *A. atra* from Otway Bay, Magallanes [79]. Intensification of LST outbreaks increased awareness of *P. reticulatum* as a primary source of YTXs and LST impacts in Magallanes. During the multi-specific blooms in 2012–2013, YTX and 45-OH-YTX (9–517 µg/kg) were among the frequently found toxins in mussels’ digestive glands. YTX was found in LC-MS analyses of picked cells and in phytoplankton samples rich in *P. reticulatum* [42]. The toxins occurred in three areas, North Magallanes (49–52° S) with YTX predominant, Central Magallanes (53–54° S) where five toxin groups were co-dominant, and South Magallanes (55° S) with PTX2 and SPXs most abundant. The highest toxin content in molluscs, mainly YTXs, was found in Central Magallanes.

#### 4.2.2. Other YTX Producers

Different species of *Gonyaulax*, i.e., *G. polygramma*, *G. spinifera*, were recorded in Northern Chile before their production of yessotoxins was known [69,102]. In March 2009, *G. taylorii* bloomed in Bahía Mejillones (23°06′ S) Antofagasta (max. 64,200 cells L^−1^. LC-MS/MS analyses revealed the presence of YTX and homo-YTX in plankton samples, with an estimated toxin content below 1 pg cell^-1^ [7]. The identification of *G. taylorii* as a new producer of YTX suggested that other *Gonyaulax* species might produce YTXs.

In northern Chile, YTXs have been associated with mass stranding of shellfish and echinoderm species [103]. In late January 2019, a mass mortality of the starfish *Stichaster striatus*, the red sea urchin *Loxechinus albus*, and the clam *Ameghinomya antiqua* were detected in Pabellón de Pica, Tarapacá (20°17′ S). Toxin analyses by LC-HRMS of digestive tissues of the affected species showed YTX ranging between 0.1 and 0.4 mg YTX kg^−1^. Two weeks later, a mass mortality of the Humboldt squid (*Dosidicus gigas*) was reported in Bahía Inglesa, Atacama (27°07′ S). About 15 t of dying or dead squids were found stranded or floating nearshore at a tourist beach in Northern Chile. LC-HRMS analyses of the visceral mass of individuals collected on 11th February revealed YTX levels of 0.42 mg YTX kg^−1^. At the end of summer, another mortality of *Do. gigas* occurred in Puerto Aldea, Coquimbo (30° S). About 130 t of these squids were collected and sold to fish processing plants and in the local market. LC-HRMS analyses revealed 0.12 mg YTX kg^−1^ in the viscera.

There are earlier reports of YTX and invertebrate mortalities, including sea urchins, starfish and abalone, from Sonoma County, California (38°30′ N). In all cases, YTX levels were below 0.1 mg kg^−1^ [104,105]. More recently, in 2017, a mortality of 250 t of abalone *Haliotis midae* in aquaculture farms in South African was associated with a bloom of *G. spinifera* [106]. The toxin profile in the digestive glands of dead specimens was dominated by homo-YTX, 45-hydroxy-YTX, and a minor contribution of YTX, with average concentrations of 0.73; 0.21 and 0.09 mg kg^−1^, respectively. Gills, with average concentration of 1.1 mg kg^−1^ (homo-YTX), 0.33 mg kg^−1^ (45-hydroxy-YTX), and 0.11 mg kg^−1^ (YTX) were the most affected organ. These levels of YTXs are comparable to those observed during the Chilean 2019 mortality event, suggesting they might have the same causative agent. However, more research is needed to determine the mechanism of action of YTXs on different tissues of marine organisms.

### 4.3. Azadinium, Amphidoma and Azaspiracids

To date, *Azadinium poporum* is the only known AZA producer reported from Northern Chile. A population of *A. poporum* near Bahía Chañaral (Atacama) reached 6840 cells L^−1^. The toxic profile consisted of AZA11, two phosphorylated derivatives, and AZA62 [8,81].

Azaspiracids have been reported in bivalves from bays including Bahía Coquimbo [82], Bahía Inglesa, and Bahía Tongoy [107] between Atacama and Coquimbo regions. However, toxin profiles, with AZA 1, and in some cases, AZA2, did not coincide with those reported for *A. poporum*. The most likely explanation is that other AZA-producing species of *Azadinium* and *Amphidoma*, are present in the area, and that AZA are widely distributed on the Chilean coast.

*Azadinium* and *Amphidoma* species are always present in plankton samples in Magallanes, but their contribution to LST events is masked in MBA tests. During February 2017, there was a mortality of Atlantic salmon being transported in mobile cages through the Gulf of Penas towards Magallanes region [19]. Local fishermen reported massive deaths of benthos, including sea urchins (*Loxechinus albus*), black snails (*Tegula atra*), piquilhue snail (*Adelomelon ancilla*), “locos” (*Concholepas concholepas*), limpets (*Fisurella* sp.), picorocos (*Austromegabalanus psittacus*), chitons (*Chiton* sp.) and macroalgae (*Macrocystis pyrifera*), in Canal Trinidad (49°60′ S; 74°53′ O) and oceanic waters off Magallanes [17].

Although *Karenia* spp. were identified as the causative organisms, *Azadinium* and *Amphidoma* species were added to the list of suspected fish killers. The hypothesis that co-occurring AZAs could have had a synergistic effect with other LST producers, was based on observations during a cruise carried out at the same time in oceanic waters. *Azadinium* and *Amphidoma* species were found in much higher densities in open Pacific waters than within the fjord channels [18]. These microalgae persisted during the year inside the channels in low numbers and reappeared in summer 2018 in oceanic waters and in interior Magallanes channels, but without reaching the high values of 2017 [19].

Additional research is needed to determine the potential sources of AZA and their possible deleterious effects on fisheries and aquaculture activities in Southern Chile.

### 4.4. Emerging Lipophilic Toxins and Their Producers

The term “emerging marine toxins” has been applied by European regulatory bodies, such as the European Commission (EC) and the European Food Safety Authority (EFSA) to designate recently described marine toxins (e.g., some cyclic imines). “Emerging marine toxins” has also been applied to non-regulated well known marine toxins apparently spreading to new areas (e.g., ciguatoxins in Atlantic subtropical areas), as well as those considered a matter of concern, requiring new toxicological evaluation (e.g., palytoxins and brevetoxins) [108].

Some cyclic imines are known to be present in Chile (Figure 4). Gymnodimine A has been found in plankton net tows by Trefault et al. [109]. Spirolides (13 desmethyl spirolide C) [82] and pinnatoxin G [110] have been found in molluscs. The producer species, however, have not been identified. *Alexandrium ostenfeldii* has been shown to produce spirolides in Perú [111], and in the Beagle Channel [112], but some Chilean strains, isolated from Aysén, did not produce this kind of toxin [86]. The only known pinnatoxin producer described up to now is *Vulcanodinium rugosum*, which has not been found in Chile. *Karlodinium* was reported in Magallanes, but the isolated strains do no produce gymnodimines or brevetoxins, only brevenal, a compound related to brevetoxins [113]. Organisms that produce palytoxins (*Ostreopsis* spp.) and ciguatoxins (*Gambierdiscus* sp.) have been identified, and some cultures established; their toxin production is still under study (G. Álvarez, unpubl. information).

### 4.5. Benthic Primary Sources of LST: Epibenthic Prorocentrum Species

Studies of benthic dinoflagellates in Chile are very scarce. Most of them are related to *Prorocentrum lima*, but there are no conclusive studies demonstrating the contribution of benthic dinoflagellates to shellfish toxicity.

A bloom of *P. lima* was reported from Bahía Calderilla, Coquimbo, during summer 2016. This little bay is located near Bahía Inglesa, the main site of scallop culture in northern Chile. Scattered *P. lima* cells were found in the water column samples and in sandy sediments, but a large proportion were epiphytic on different macroalgae, such as *Codium fragile*, *Agarophyton chilensis*, and *Ulva* sp. Cells were isolated, culture established, and their toxin profile found to consist of OA (3.53 pg cell^−1^), followed by DTX1 (0.012 pg cell^−1^) and traces of C8 and C10 OA diol-esters [114].

More recently, Álvarez et al. [115] reported the presence of toxic benthic dinoflagellates *Prorocentrum rhathymum* and *Prorocentrum* cf *consutum* from Easter Island (Rapa Nui). Strains of both species were isolated and established in culture. UHPLC-HRMS analyses showed that *P. rhathymum* has a toxin profile dominated by OA (0.983 pg cell^−1^) followed by epi OA (0.230 pg cell^−1^) and DTX2 (0.025 pg cell^−1^), whereas that of *P.* cf. *consutum* is dominated by OA (0.7 pg cell^−1^) with traces of epi OA and DTX2. DTX1 was not detected in either species.

In Magallanes, *P. lima* strains were reported to contain OA and DTX1 [116]. Empty cells of *P. lima* have been found in the gut contents of various browsing gastropods, e.g., limpets [117]. Although development of *P. lima* blooms is unlikely, this information should be considered in situations where okadaates of unknown origin are found in benthic resources. It may also help reveal different vectors and pathways of the toxins and potential transfers from benthic to planktonic food webs [79].

## 5. Lipophilic Shellfish Toxins and Their Accumulation and Biotransformation by Bivalve Molluscs in Chile

Lipophilic toxins are accumulated by bivalves along the entire Chilean coast. Their impact, however, varies considerably between regions according to the toxic potential of the local strains and the biotransformation of their toxins by the locally exploited bivalve species. The most outstanding differences between strains are those observed in *D. acuminata*: those potentially producing DTX1 in addition to pectenotoxins are far more damaging for public health (gastrointestinal syndrome) and the shellfish industry.

On the northern coast, the most important bivalve species are the scallop *Ar. purpuratus* and the clams *Me. donacium* and *Mulinia edulis* (Table 1). *Dinophysis* species in this region seem to produce nearly exclusively pectenotoxins, mainly PTX2 [54,55,56]. OA and DTXs are rarely detected and then in very low concentration. Up to now, the maximum PTX2 level found in *Ar. purpuratus* barely reached the RL, i.e., 160 µg kg^−1^, which is consistent with some market closures imposed using the MBA to quantify DSP toxicity. In *Me. Donacium*, the maximum concentration found was 120 µg kg^−1^ [56], with a much higher concentration of PTX2-sa (PTX2 seco-acid), which is a non-toxic product from bivalve biodegradation of this toxin. Only one value (3.64 µg kg^−1^) of PTX2 in *M. edulis* is available [54]. A sample of *Me. donacium* taken from the same area collected at the same time had a slightly higher concentration (4.64 µg kg^−1^) of PTX2 and a lower amount of its degradation product, PTX2sa. These results suggest that the clam Me. donacium accumulates higher levels of PTXs than the clam *Mu. edulis*. PTX2 depuration rate has been estimated only for *Me. donacium* [56]. The apparent depuration rate, i.e., the combination of direct elimination and transformation to PTX2-sa, was 0.3 d^−1^, which corresponds to a semi-depuration time of 2.3 d. The high depuration rates found make the accumulation of large amounts of this toxin by *Me. donacium* highly unlikely.

On the southern coast, *Dinophysis* toxin profiles include PTXs and okadaates (DTX-1 and OA). The main species affected are mytilids *My. chilensis* and *Au. atra*, which inhabit rocky shores, and clams *Gari solida* and *Ameghinomya antiqua*, which live in soft bottoms. Maximal total concentrations of DTX-1 of 390 and 266 µg kg^−1^ in *Au. atra* and *My. chilensis*, respectively [118,119], and 206 and 168 µg kg^-1^ in *G. solida* and *Am. antiqua* [120] have been found. The apparent depuration rates of DTX-1 from *Au. atra* and *My. chilensis* (computed from plots shown in García-Mansilla [119], and from analyses of a combination of the two species in Alves de Souza et al. [52]) ranged from 0.02 to 0.18 d^−1^ and averaged 0.10 d^−1^, which corresponds to a semi-depuration time of 7 d. In soft bottom species (*G. solida* and *Am. antiqua*, undifferentiated) depuration seems to be faster, with an estimated average rate of 0.15 d^−1^, which is equivalent to a semi-depuration time of 4.6 d.

Yessotoxins, first reported (but not quantified) in 1997 in mussels from the Chonos Archipelago [35], have been detected in plankton concentrates and shellfish along the entire Chilean coast from Arica to Magallanes. The maximal concentration of YTX recorded to date was approximately 129 µg kg^−1^ (517 µg kg^−1^ of YTX + 45OH-YTX in the digestive gland) in *My. chilensis* collected from Magallanes [43]. The estimated depuration rates of these toxins in bivalves from Los Lagos and Aysén are quite high, ranging from 0.1 to 0.26 d^−1^, and averaging 0.17 d^−1^ which corresponds to a 4-d semi-depuration period.

Low concentrations of AZAs, SPXs, and PnTXs have also been found in Chilean molluscs [82,83,121]. There is no information about the cellular content of these toxins in dinoflagellate strains from the area. A strain of *A. ostenfeldii* isolated from the Beagle Channel, Argentina, (near Magallanes) was shown to contain a low amount of 20-methyl SPX-G (0.15 pg cell^−1^) and 13-desmethyl SPX-C (0.59 pg cell^−1^) [112]. Consequently, if the Chilean strains have a similar cell content, the bivalves which feed on this species would not accumulate large amounts of SPXs unless the depuration rate was extremely low.

Filter-feeding bivalves partially biotransform the toxins contained in the plankton they feed upon. In the clam *Me. donacium*, PTX2 is transformed to its seco-acid, PTX2sa, and this in turn to different acyl-derivatives by esterification with fatty acids. Hydrolysis of PTX2 to its seco-acid derivative takes place in other bivalve species in Chile, including *Ar. purpuratus*, *Semimytilus algosus*, *Mu. edulis* [54], *My. chilensis*, *Am. antiqua* [122], *Au. atra* [118,119] and *T. dombeii* [51]. This transformation is common to other bivalve species from different geographical areas [123,124,125,126,127]. In addition, clams *T. dombeii*, from Bío Bío region seem to produce an unidentified PTX2sa isomer [51]. These biotransformation products are not toxic to mice by intraperitoneal injection.

The rate of transformation is not the same in all species. *My. chilensis*, for example, hydrolyses PTX2 to PTX2sa faster than *Am. antiqua* [122]. OA and DTX1 seem to be transformed in Chilean bivalves to 7-*O*-acyl esters of fatty acids (generically known as DTX3), as the amount of free toxins usually increases following alkaline hydrolysis of the shellfish extracts. These acylation reactions take place in all studied bivalves, and they are probably an important step in the depuration process of these toxins [128] (Figure 11). As in other parts of the world, including Spain, the acylation capability varies within bivalve species [129]. The Chilean mussels. *M. chilensis*, for example, seem to esterify DTX1 at a higher rate than *Au. atra*. This suspicion is based on observations of the lower percentages (38.5 vs. 55% on average) of free toxin found in several sympatric populations [118]. Esterification rates in *M. chilensis* are also faster than in the clam *Am. antiqua* [122].

Recently, *D. acuta* cells from Puyuhuapi Fjord, Aysén, were found with OA and DTX1 in conjugated form. This implies that diol-, triol- esters or even more complex compounds (e.g., DTX4 or DTX5, Figure 1) can be ingested by bivalves. These complex compounds may be totally or partially hydrolysed in bivalve digestive systems by hydrolases in the phytoplankton cells [130], or even by specific enzymes, such as that described by MacKenzie et al. [131]. When partially hydrolysed, hybrid esters in which the carboxylic function of the toxin esterifies a diol or triol, and a fatty acid esterifies the C-7 hydroxyl, can also be generated [132]. These esters have not so far been reported from Chilean bivalves.

Depuration rates are affected by environmental conditions at aquaculture sites. For example, depuration rates recorded in mussels from Chiloé and Galicia were much higher than rates found for the blue mussel, *My. edulis* in Bjorsund (~58° N), Norway [133]. These differences are probably due to temperature, a relationship well demonstrated for the clam *Donax trunculus* [134]. The lower Norwegian rates can probably be applied to Magallanes (~48 to 56° S), where environmental conditions are similar to the Norwegian fjords.

## 6. New European Regulations for Lipophilic Toxins: Impacts on the Chilean Shellfish Industry, Artisanal Fisheries and Other Coastal Commodities

The exponential development of the Chilean shellfish industry in the present century shares many similarities with that of the Galician mussel industry, northwest Spain, from the 1950’s to the 1970’s of the last century. In Galicia, the Mediterranean mussel (*Mytilus galloprovincialis*) industry developed in the Rías Baixas, with a latitudinal extent of about 100 km. The Chilean mussel *My. chilensis* (“chorito”) industry grew, concentrated on the eastern side of the 180 km long Chiloé. Both regions had to face the common threat posed by various kinds of harmful algal blooms and the market disruptions caused by sanitary harvesting bans.

Nevertheless, the impact of LST is substantially different in the two regions. Thus, the incidence of lipophilic toxins on bivalves, and in particular on cultured mussels, seems to be much lower in Chile than in Galicia. For example, estimates of OA and DTXs prevalence in mussels in Chiloé from 2015 to 2018 (0.1%) [47] are much lower than those in a normal year in Galicia (~71.1%) [129]. About 10.3% of the latter (71% Galician positive samples) surpassed R.L. for DSP toxins in the same period of time. In contrast, none of the Chiloé samples where LST were present (>LOD) had DSP toxins exceeding the R.L. As a result, there were no harvesting bans in Chiloé associated with *D. acuminata*, whereas the same species has been the cause of more than 9 month closures, as in 2007 at the Galician DSP hot spot in Ria de Pontevedra [12]. These differences cannot be explained on the basis of observed depuration rates, which are similar in the two places [129,135,136]. The same, to a lesser extent, is the case with estimates of YTX prevalence, 7.1% in Chiloé and 10.4% in Galicia.

This is not the case with PTXs, which had a much higher prevalence in Chiloé than in Galicia (1.6 and 0.2%, respectively) from 2015 to 2018. In both regions *D. acuminata*, with a very long growing season (spring to autumn) is the predominant *Dinophysis* species; but they have very different toxin profiles. Galician strains contain only OA (up to 40 pg·cell^−1^), whereas Chilean strains apparently containing only moderate to low levels of PTX2 were predominant in Chiloé in the 2015–2018 period.

There is increasing evidence that pectenotoxins do not pose a risk to human health [137,138]. In the light of the most recent toxicological studies, the European Commission decided to deregulate these compounds. This change is advantageous since it reduces the amount of products with toxin content above RL, and days of harvesting bans of shellfish species accumulating PTXs. Nevertheless, deregulation of PTX2 in Chile has very different impacts on different seafood resources depending on whether their destination is the national market or exports to European countries. In northern Chile, regulatory changes on PTXs probably will not imply substantial changes to the growing *Ar. purpuratus* exports for several reasons. First and most important, the toxin control of scallop exports in Chile benefits from the exception of EC directives affecting control of amnesic shellfish poisoning (ASP) toxins in pectinid bivalves, i.e., only the adductor muscle and the gonad are to be analysed [139]. This control excludes the digestive gland, which is the organ that contains the highest proportion of lipophilic toxins, and consequently concentrations of any other kind of toxins in addition to ASP are greatly reduced. So far, LST levels in scallops from northern Chile have never surpassed the regulatory limit, and only a reduced number of market closures have affected other commercial species from the region.

Positive impacts will also be very limited for the local artisanal fisheries. Even if local regulations were modified to deregulate PTXs, maintenance of the MBA for toxicity testing would make it impossible to distinguish PTXs from the diarrhetic compounds of the OA group. Nevertheless, if changes in toxin determination methods were added to those in regulated toxins, days of harvesting closures in the northern “OA-free regions” would be considerably reduced or even disappear. These changes would have an important positive impact on local fishing communities in northern Chile. These communities suffer from long unnecessary harvesting bans as a consequence of the low (biweekly or monthly) frequency of inadequate toxin tests and the lack of monitoring of potentially harmful microalgae species.

The situation for artisanal fisheries in Chilean Patagonia is not much better. Unlike the northern regions, there are fixed stations for phytoplankton monitoring that complement toxin tests and help decision makers organize contingency plans more efficiently. Nevertheless, stakes are high for many small companies relying on MBA to ensure seafood safety in the south. Very complex multi-specific events often occur in those regions where already deregulated (PTXs) and non-regulated emerging toxins (e.g., SPXs) co-occur and may cause unnecessary harvesting bans.

## 7. Conclusions and Future Perspectives

Moderate levels of lipophilic toxins are accumulated in shellfish along the entire Chilean coast, but the associated socioeconomic impacts are determined, to a large extent, by the strain-specific toxin profiles of the phytoplankton species involved and by the toxin determination method applied (MBA for the national market, LC-MS for exports) to commercial shellfish species in each region.

*Dinophysis acuminata* and *Protoceratium reticulatum*, present in coastal waters from all Chilean regions, are the more frequent and abundant toxin producers; *D. acuta*, with a high cellular content of OAs, is the most damaging and the only *Dinophysis* species associated with OA in the country. However, unambiguous information on toxin profiles by LC-MS analyses of picked cells and cultures is practically inexistent.

None of the commercially important bivalve species examined have shown long retention times for any of the accumulated lipophilic toxins. Nevertheless, low water temperatures in the fjord regions, in particular in Magallanes, may have a negative effect on depuration rates by increasing the time needed to metabolize toxins.

Cyclic imines and other emerging toxins have been detected in the area, but the primary producers have not been identified.

On the basis of this review, several issues are identified that require consideration: (i) The continuation of efforts to isolate and establish cultures of different strains of the *D. acuminata* complex from Chilean waters, characterize their toxin profiles and content and determine their distributions; (ii) The exploration of the cause of shifts and contractions of the geographical ranges of *D. acuta* populations; (iii) Blooms of YTX producers in northern Chile, and the persistence of these compounds in the southern regions, require studies to determine their negative effects on commercial species, and their possible contribution to recent shellfish strandings. However, the most urgent issue is to revise ongoing monitoring systems and resolve the issue of their low temporal resolution, aggravated in the northern regions, which lack phytoplankton monitoring. More frequent LC-MS analyses to solve uncertainties about positive mouse bioassays after intraperitoneal injection of lipophilic shellfish extracts, and rapid tests for toxins screening in locations of difficult access should be considered.

## Figures and Tables

**Figure 1 marinedrugs-20-00122-f001:**
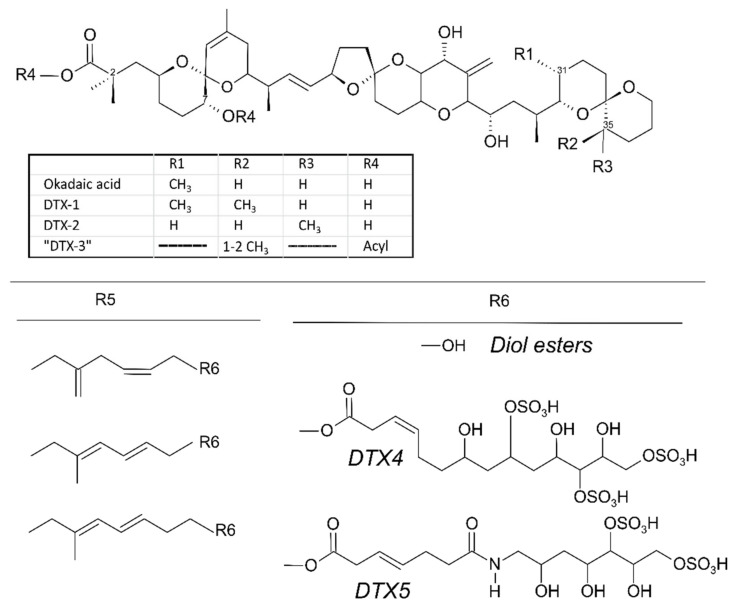
Structure of okadaic acid (OA), dinophysistoxins (DTXs) and their derivatives. DTX3 is produced by metabolic acylation of the main toxins in bivalves. Diol esters, DTX4 and DTX5 are produced in phytoplankton by esterification of different chains with the carboxylic function of the main toxins.

**Figure 2 marinedrugs-20-00122-f002:**
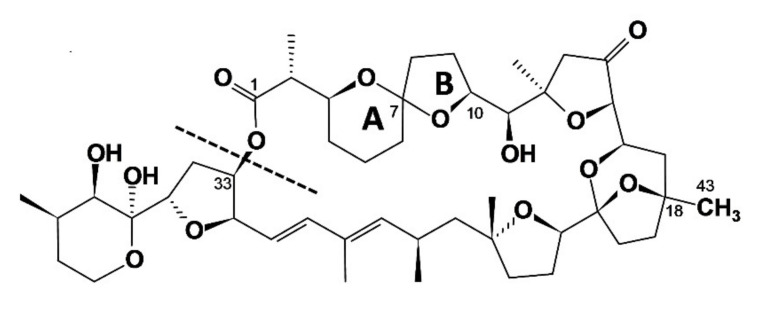
Structure of Pectenotoxin 2 (PTX2), the main PTX found in Chilean phytoplankton. The dotted line shows the location where the macrocycle opens by hydrolysis to generate the PTX2 seco-acid (PTX2sa.), main PTX2 biotransformation product found in Chilean shellfish.

**Figure 3 marinedrugs-20-00122-f003:**
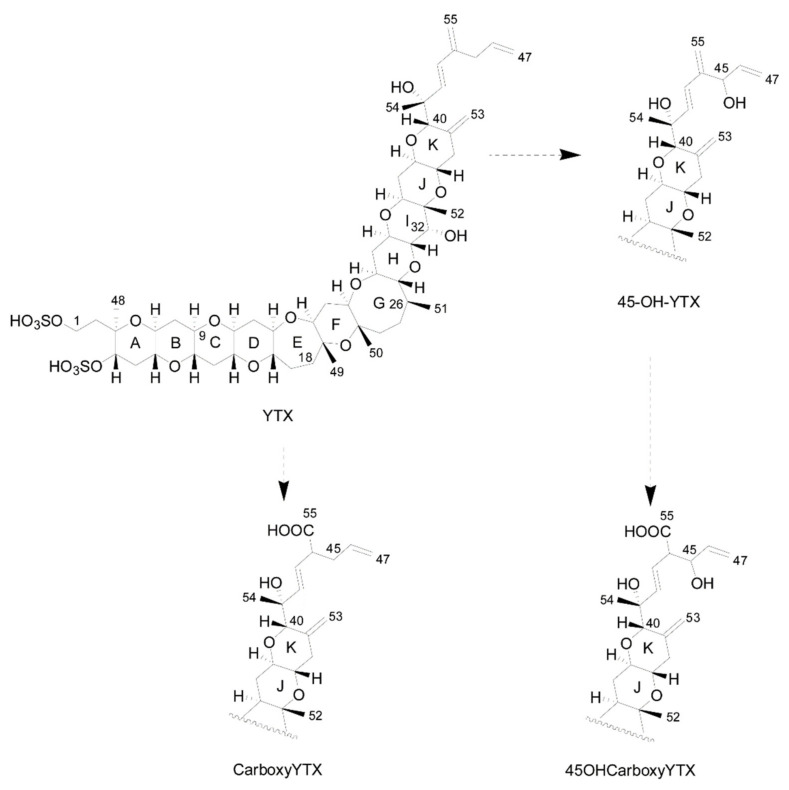
Structure of the main yessotoxins and routes of biotransformation. From Paz et al. [9].

**Figure 4 marinedrugs-20-00122-f004:**
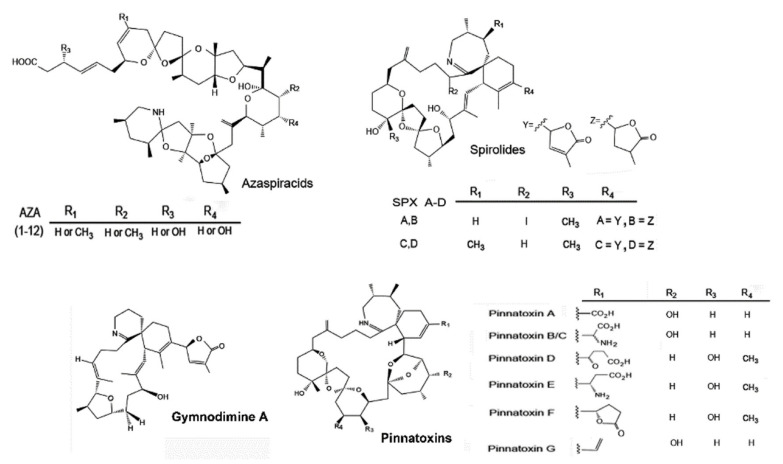
Structures of the main azaspiracids (AZA), spirolides (SPX), pinnatoxins (PnTX), and gym-nodimine (GYM) found in Chilean waters. Partially redrawn from Leyva-Valencia et al. [14].

**Figure 5 marinedrugs-20-00122-f005:**
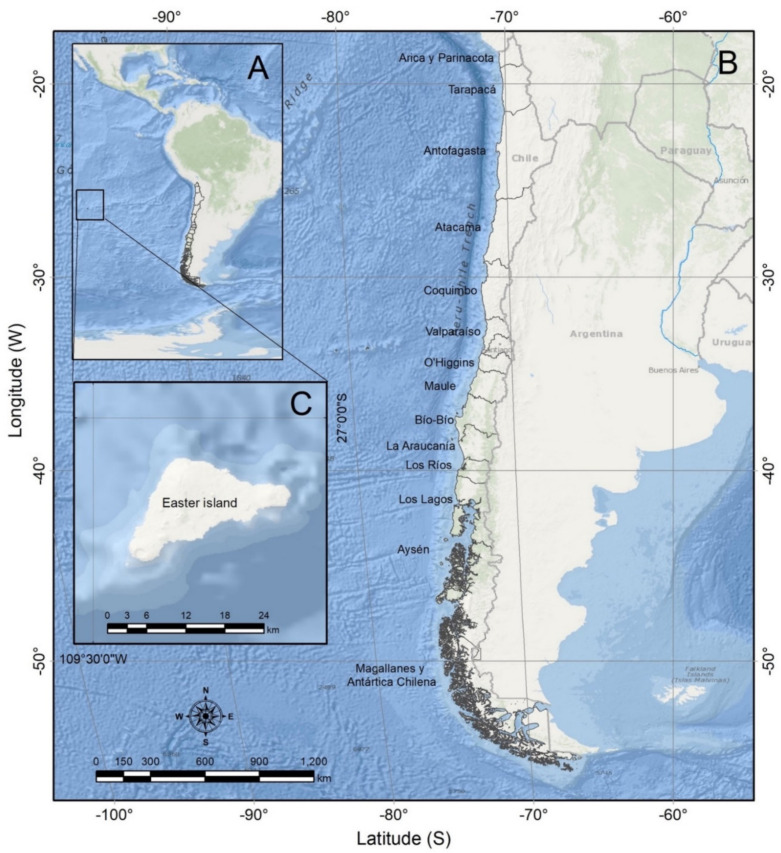
Map showing: (**A**) South America; (**B**) Chile; (**C**) Easter Island (Rapa Nui). The names of administrative regions are shown to the right of each region.

**Figure 6 marinedrugs-20-00122-f006:**
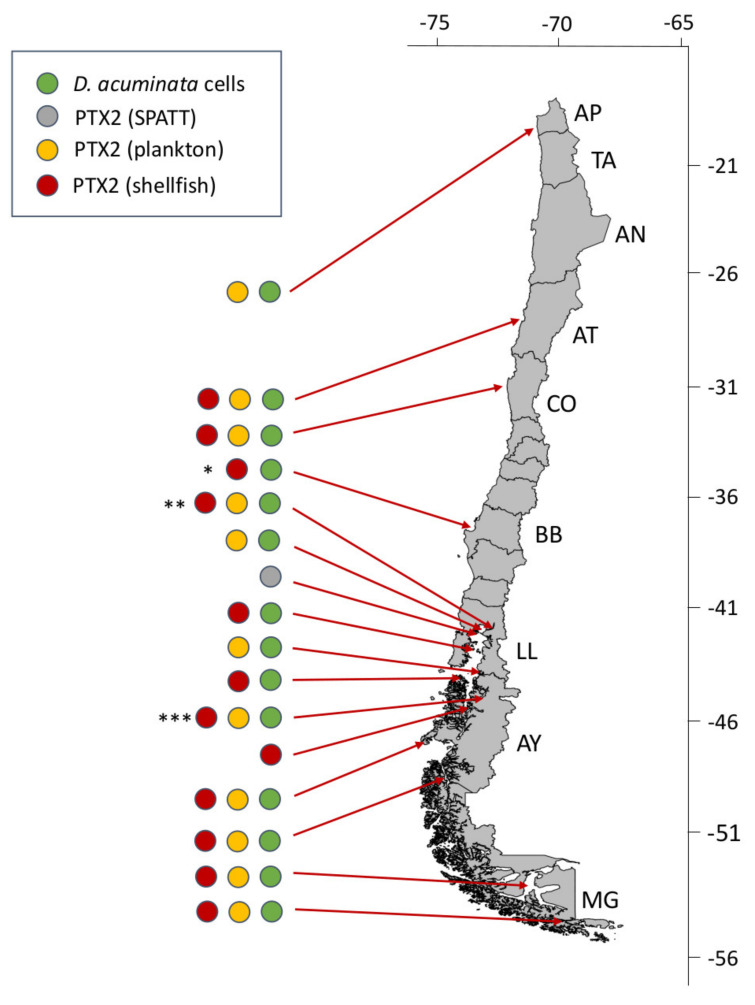
Latitudinal distribution of historic pectenotoxins-2 (PTX2) events in Chile associated with *Dinophysis acuminata*. Asterisks indicate sites of cell maxima (integrated 0–10 or 0–15 m tube samples) during the three most severe events in Bío Bío (* max. 39,520 cells L^−1^), Los Lagos (** max. 11,300 cells L^−1^) and Aysén (*** max. 6,800 cells L^−1^) regions in spring 2019 and summers 2008 and 2019, respectively.

**Figure 7 marinedrugs-20-00122-f007:**
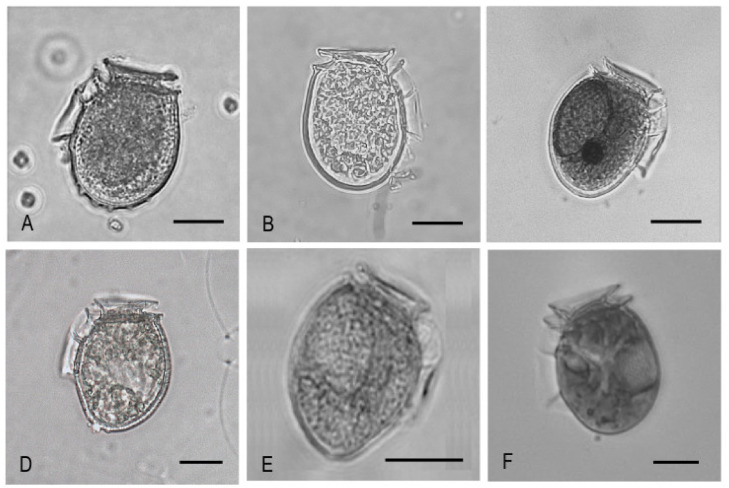
Light microscopy micrographs of *Dinophysis acuminata* complex morphotypes from different regions of the Chilean coast: (**A**) Coquimbo; (**B**) Bío Bío; (**C**) Los Lagos; (**D**) Aysén and (**E**,**F**) Magallanes. Scale bar = 20 µm. (Micrographs (**E**,**F**) courtesy of Hernán Pacheco).

**Figure 8 marinedrugs-20-00122-f008:**
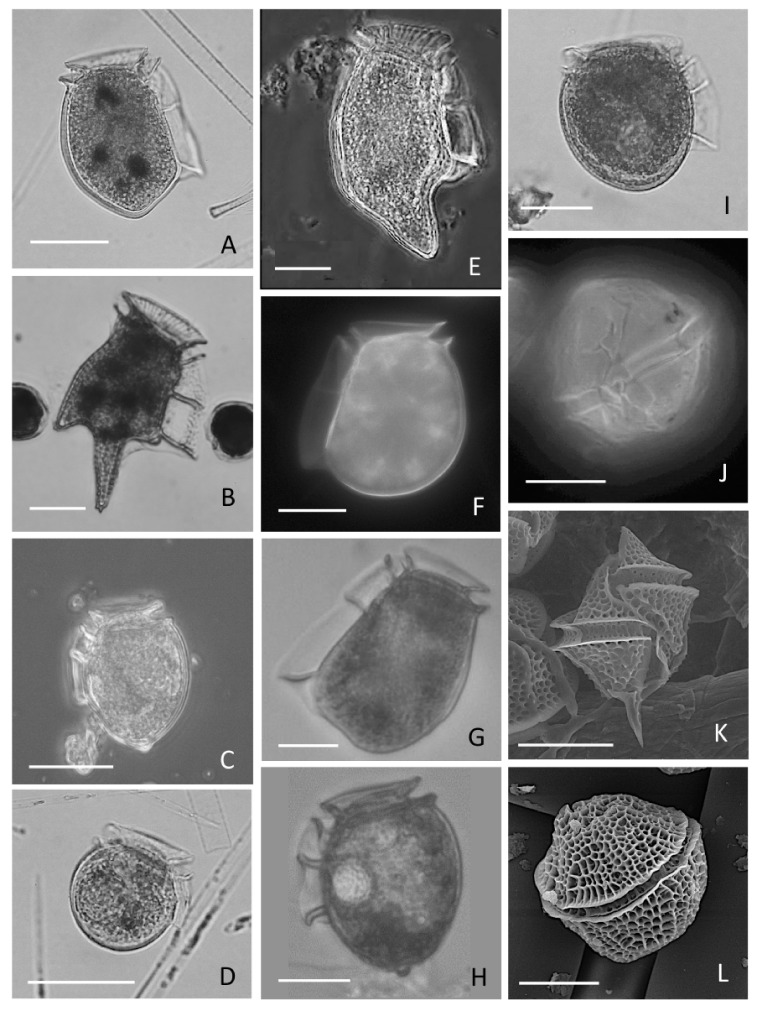
Light (**A**–**J**) and scanning electron microscopy (**K**–**L**) micrographs of potential producers of lipophilic toxins in Chile. (**A**) *D. acuta*; (**B**) *D. tripos*; (**C**) *D.* cf *norvegica*; (**D**) *D. subcircularis*; (**E**) *D. caudata*; (**F**) *D. fortii*; (**G**) *D. truncata*; (**H**) *Dinophysis* sp.; (**I**) *Phalacroma rotundatum*; (**J**) *Alexandium ostenfeldii*; (**K**) *Gonyaulax spinifera*; (**L**) *Protoceratium reticulatum*; Scale bar = 20 µm. (Micrograph (**F**) courtesy of Verónica Muñoz and (**C**) of Hernán Pacheco).

**Figure 9 marinedrugs-20-00122-f009:**
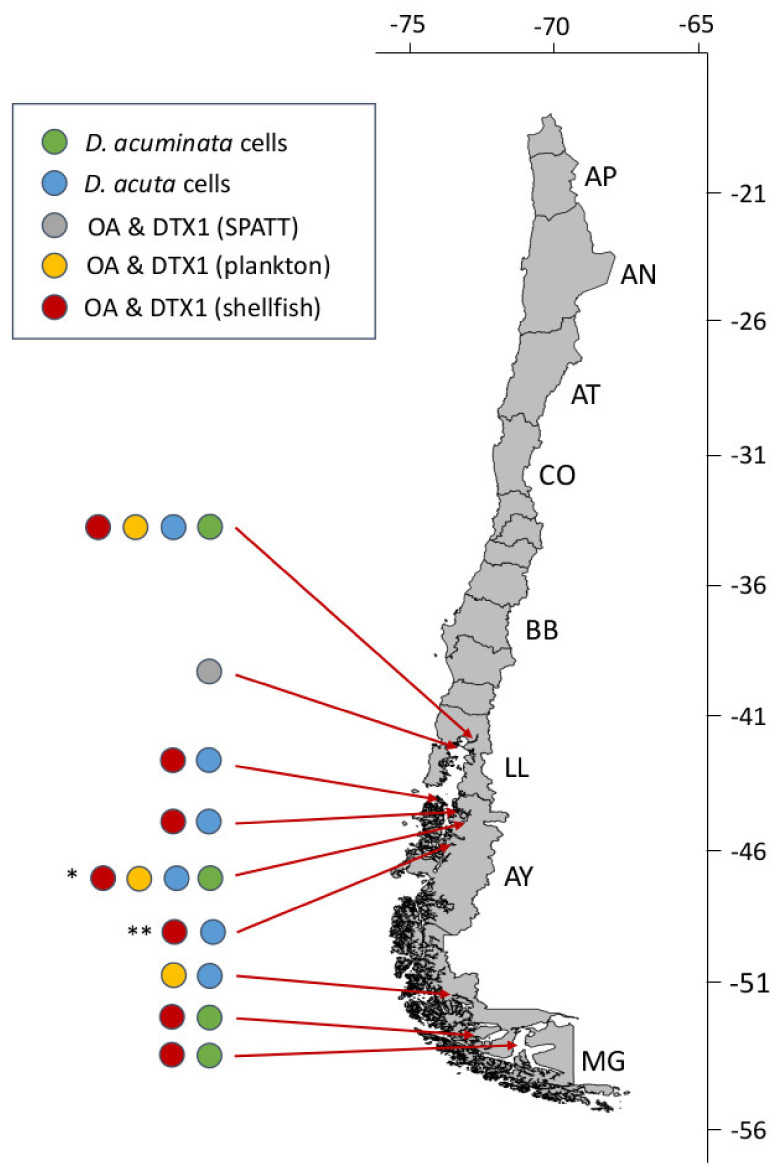
Latitudinal distribution of historic diarrhetic shellfish poisoning (DSP) outbreaks in Chile, associated with *D. acuta* and *D. acuminata*. Asterisks indicate sites where cell maxima occurred during the two most severe events of *D. acuta* in Aysén region in summer 2018 (* 664,000 cells L^−1^) and 1991 (** 7000 cells L^−1^).

**Figure 10 marinedrugs-20-00122-f010:**
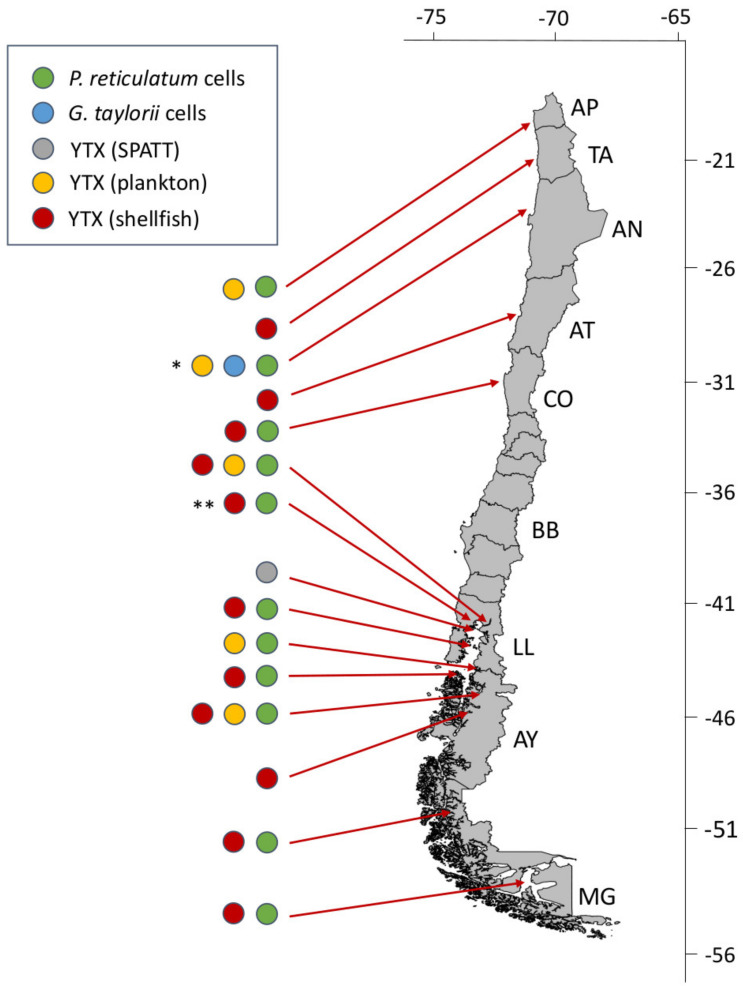
Latitudinal distribution of historic yessotoxins (YTX) events associated with *Protoceratium reticulatum* and *Gonyaulax taylorii*. Asterisks indicate sites where cell maxima occurred during the two most severe events of *P. reticulatum* in Antofagasta (* max. 383,520 cells L^−1^) and Los Lagos (** max. 12,000 cells L^−1^) regions in summers 2007 and 2015, respectively.

**Figure 11 marinedrugs-20-00122-f011:**
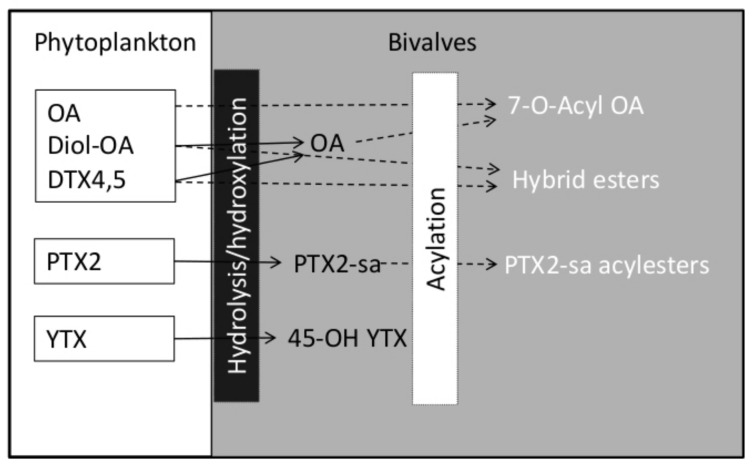
Biotransformation of the most important lipophilic toxins in shellfish.

**Table 1 marinedrugs-20-00122-t001:** Landings (tons) of the main commercially exploited bivalve species from natural shellfish banks (artisanal fisheries) and from aquaculture along the Chilean coast during 2020. Codes for regions: CO: Coquimbo; BB: Bío Bío; LL; Los Lagos; AY: Aysén; MG: Magallanes; OR: Other 10 regions. Data from SERNAPESCA 2020 [49].

Scientific Names	Common Names (English)	Common Chilean Names (Spanish)	Price ($US kg^−1^)	Administrative Regions						Total
				CO	BB	LL	AY	MG	OR	
*Mytilus chilensis*	Blue mussel	Chorito	1.0		7	400,230	4	28	35	400,304
*Ameghinomya antiqua*	Clam	Almeja	0.8	1	92	10,993	57	12	133	11,288
*Aulacomya atra*	Ribbed mussel	Cholga	0.6		109	6037	32	55	81	6314
*Argopecten purpuratus*	Scallop	Ostion del Norte	14,5	3987					380	4367
*Tagelus dombeii*	Hard razor clam	Navajuela	0.7		2700	221			249	3170
*Choromytilus chorus*	Giant mussel	Choro	0.5		41	2172	99		281	2593
*Tawera gayi*	Baby clam	Juliana	0.2			2504			0	2504
*Mulinia edulis*	Clam	Taquilla	0.1		1191				0	1191
*Ensis macha*	Sea asparagus	Huepo	1.3		832	328			0	1160
*Mesodesma donacium*	Surf clam	Macha	5	780		71			4	855
*Chlamys vitrea*	Scallop	Ostion del Sur	0.8				1	446	0	447
*Ostrea chilensis*	Chilean oyster	Ostra Chilena	0.9			401			0	401
*Gari solida*	Clam	Culengue	1.0	17		282	47		0	346
*Semele solida*	Clam	Tumbao	0.6			159			0	159
*Magallana gigas*	Pacific oyster	Ostra del Pacifico	6.2	67	2	23			2	94
*Prothotaca taca*	Clam	Taca	1.4	2					0	2
Total				4853	4882	412,428	183	529	1032	423,907

**Table 2 marinedrugs-20-00122-t002:** Main putative primary producers of lipophilic toxins recorded along the Chilean coast. Codes for regions: AP: Arica y Parinacota; TA: Tarapacá; AN: Antofagasta; AT: Atacama; CO: Coquimbo; VA: Valparaíso; OH: O’Higgins; ML: Maule; ÑU: Ñuble; BB: Bío Bío; AR: Araucanía; LR: Los Ríos; LL; Los Lagos; AY: Aysén; MG: Magallanes. Bold denotes toxicity confirmed by LC-MS.

	Species		Administrative Regions													References
		AP	TA	AN	AT	CO	VA	OH	MA	BB	AR	LR	LL	AY	MG	
DSP/PTX	** *Dinophysis acuminata* **	+	+	+	+	+	+	+	+	+	+	+	+	+	+	[13,24,42,47,50,51,52,53,54,55,56,57,58,59,60,61]
toxins	** *D. acuta* **												+	+	+	[31,32,53,61,62,63,64,65]
	*D. caudata*	+	+	+	+	+							+			[23,64,66,67]
	*D. forti*	+	+	+						+						[23,24,51,67]
	*D. norvegica*														+	[59]
	*D. ovum*	+	+	+	+	+										[24,68]
	*D. saculus*		+	+	+	+										[24,69]
	*D. tripos*	+											+			[64,68]
	*Phalacroma rotundatum*	+	+	+	+	+	+	+	+	+	+	+	+	+	+	[55,70]
	*D. exigua*			+	+	+										[24]
Potential	*D. hastata*	+	+													[23,24,67]
DSP/PTX	*D. schuettii*	+	+													[23,24,67]
	*D. subcircularis*												+	+		[52,64]
	*Phalacroma rapa*			+	+	+										[24]
YTX	** *Protoceratium reticulatum* **	+	+	+	+	+	+	+	+	+	+	+	+	+	+	[36,47,52,55,71,72,73,74,75,76,77,78,79]
	*Gonyaulax spinifera*	+	+	+	+	+	+									[80]
	** *G. taylorii* **			+	+	+	+									[7]
AZP	*Azadinium poporum*		+	+											+	[8,17,18,19,81,82,83]
	*Amphidoma* spp.														+	[17,18,19,34]
SPX	*Alexandrium ostenfeldii*				+	+					+			+		[84,85,86]

## Data Availability

Not applicable.

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
