# Peer review of "Lipophilic Toxins in Chile: History, Producers and Impacts"

_marinedrugs, 2022, doi:10.3390/md20020122_

Round 1

Reviewer 1 Report

Report on the manuscript entitled (Lipophilic Toxins in Chile: History, Producers and Impacts) submitted Marine Drugs is a quite popular research, have been dealt by many authors previously. However, there is always a scope to review and crystallize new information for overall benefaction of the scientific society working in this direction. Therefore, this manuscript requires minor corrections then it will be re-submitted to the Marine Drugs.

  • The topic is timely and worth of revision. However, the paper lacks a coherent structure and in its present form is merely a compilation of information without liaison and critical analyses.
  • Abstract: needs more quantitatively
  • lack of recent literature (Recent references (last 3 years) should be included
  • The English language should be revised by expert
  • The review is well written and organized. In spite of its local nature, but it is important in the toxicology of marine habitats.
  • The plagiarism is 11%, which is very good. English needs moderate revision in some places of the MS

Author Response

Reviewer 1

Report on the manuscript entitled (Lipophilic Toxins in Chile: History, Producers and Impacts) submitted Marine Drugs is a quite popular research, have been dealt by many authors previously. However, there is always a scope to review and crystallize new information for overall benefaction of the scientific society working in this direction. Therefore, this manuscript requires minor corrections then it will be re-submitted to the Marine Drugs.

  • The topic is timely and worth of revision. However, the paper lacks a coherent structure and in its present form is merely a compilation of information without liaison and critical analyses.

Authors: This comment (“lacks a coherent structure”) seems to be in contradiction with what the same reviewer wrote in his comment before last (“The review is well written and organized”). We respectfully disagree about a lack of critical analyses in our review.

  • Abstract: needs more quantitatively

Authors: It is difficult to be certain of what the reviewer’s demand is. Maybe it refers to have quantitative figures in the abstract? We tried our best to fit in the 200 words allowance a summary of the whole review. There was no space for details.

  • lack of recent literature (Recent references (last 3 years) should be included.

Authors: This is a review and references were chosen if they were relevant to the topic under discussion regardless of the year of publication. Concerning the last 3 years mentioned by the reviewer, we have: 7 references from 2021, 16 from 2020 and 19 from 2019

  • The English language should be revised by expert

Authors: We agree. The full manuscript has been carefully revised by an experienced scientific English editor.

  • The review is well written and organized. In spite of its local nature, but it is important in the toxicology of marine habitats.

Authors: Thanks, we appreciate this comment.

.

  • The plagiarism is 11%, which is very good. English needs moderate revision in some places of the MS

Authors: Do we have to take this as a compliment? Certainly, any sentence referring to other authors’ papers has been properly cited

Reviewer 2 Report

This is an interesting and well-structured review on the long term history of lipophilic toxins’ occurrence in Chile, discussing the producing species, bivalve molluscs and other marine organisms affected as well as the impacts thereof. There are, however, several points needing improvement in order to make the manuscript more robust and reader friendly.

General remarks:

  1. Throughout the text: some weird abbreviations are used, e.g. Mu. chilensis instead of the correct M. chilensis. The authors should use the common abbreviations for species names. Also, all species names should be abbreviated after their first instance in the text and throughout the text. There are multiple instances of writing full species names again and again, especially in the last 3 sections of the manuscript so this should be corrected throughout the text.
  2. Please check that all instances of genus and species names are italicized, there are multiple instances in the text where this is not the case.

Specific remarks:

  1. Introduction

- Page 2, lines 48-49: Species of the genus Phalacroma, producers of OA, should also be included.

- Page 2, lines 78-79: DSP/LST intoxications are either non-fatal or rarely fatal, please revise.

  1. Historical overview

- Page 4, lines 115-116: “… early shellfish poisoning outbreaks”. The authors should clarify what they mean by “early” outbreaks – does this refer to those in the very past (e.g. 60s)?

- Page 4, lines 117-119: As also mentioned later in the manuscript, Mesodinium is a ciliate and not considered as a HAB organism (or red tide one) – it is only a prey. This should be revised here by the authors.

  1. Monitoring of lipophilic shellfish toxins in Chile

- There are multiple references to regulatory limits for LSTs (e.g. for OA, YTXs, etc.) throughout the manuscript, but the readers are not informed of what is actually regulated in Chile. The legislation/regulatory limits implemented in Chile should also be discussed at this point

- Page 5, lines 171-172: Completely obsolete EU legislation is referenced, please update with current legislation in force. The particular item cited as [35] has been repelled since 2004.

- Page 5, line 209: “acquainted personnel”: knowledgeable or experienced personnel would be a better alternative.

  1. Lipophilic Shellfish Toxin producers in Chile: distribution and toxic events

- Page 8, line 316: “…DTX3 in shellfish above RLs (280 and 239 μg kg−1 HP”. No previous reference to existing RLs for Chile has been made - the authors should include this information earlier in the manuscript. Furthermore, “HP” (probably hepatopancreas” is an abbreviation not explained previously in the text. The authors should also clarify how this concentration in HP exceeds the RLs, since the latter are referring to shellfish meat (whole tissue) and not HP.

- Page 9, line 333: “…over 11,000 samples collected”: what kind of samples? it's probably shellfish, although not mentioned, as otherwise regulatory limits would not be applicable, nevertheless this should be clarified.

- Page 9, line 334: “tested positive for LST”: tested positive commonly refers to exceeding regulatory limits. It would be better to use another expression, such as confirmed to contain LSTs or equivalent.

- Page 11, line 404: “majoritarian”: Better use another term, e.g. main, major, etc.

- Page 12, line 432: “They have been always assumed to be no relevant potential contributors”: please rephrase, this is not comprehensible.

- Page 12, line 444: “20,07”: do the authors mean “2007”?

- Page 15, line 502: Please provide a reference for the EU regulatory level for YTXs.

- Page 19, lines 633-634: “This microalga that has been found in the gastric content of browsers, e.g. limpets”: please rephrase, the syntax is confusing.

  1. Lipophilic shellfish toxins and their accumulation and biotransformation by bivalve mollusks in Chile

- Page 19, line 664: “2.3 days)”: remove bracket.

- Page 20, lines 686-693: This section refers to toxins presence in bivalve molluscs. Information on toxin content of phytoplankton cells belongs elsewhere (in earlier sections of the manuscript).

  1. LST new regulations and impacts on the Chilean shellfish industry, artisanal fisheries and other coastal commodities

- Page 21, line 750: “por”: “for” maybe?

- Page 22, lines 774-778: The authors should mention these specific directives. To my knowledge, EU Decision 2002/226/EC (to which this point probably refers) deals only with domoic acid content in pectenids. This practice is not yet regulated for LSTs.

- Page 22, line 782: “MB”: “MBA” maybe?

- Page 22, lines 795-796: “toxins planned to be deregulated soon (e.g.YTXs)”: Please provide a reference for this information. Is this within the EU, Codex, Chile?

  1. Conclusions and future perspectives

- Page 22, line 807: “inambiguous”: “unambiguous” maybe?

- Page 23, line 826: “lacking a phytoplankton”: “lacking a phytoplankton testing routine” maybe?

Author Response

Reviewer 2

We appreciate the detailed revision of Reviewer 2 All his suggestions have been taken into account.

Comments and Suggestions for Authors

This is an interesting and well-structured review on the long term history of lipophilic toxins’ occurrence in Chile, discussing the producing species, bivalve molluscs and other marine organisms affected as well as the impacts thereof. There are, however, several points needing improvement in order to make the manuscript more robust and reader friendly.

General remarks:

Throughout the text: some weird abbreviations are used, e.g. Mu. Chilensis instead of the correct M. chilensis. The authors should use the common abbreviations for species names. Also, all species names should be abbreviated after their first instance in the text and throughout the text. There are multiple instances of writing full species names again and again, especially in the last 3 sections of the manuscript so this should be corrected throughout the text. Please check that all instances of genus and species names are italicized, there are multiple instances in the text where this is not the case.

Authors: We used two letters instead of one for the abbreviation of the generic name to avoid confusion when there were several genera starting with the same letter in the same paper. Some journals even recommend to do that. Nevertheless, we have changed this now to follow this reviewer’s suggestions. Concerning the full latin name of the species, we have put it the first time it appeared in the abstract, in the figure captions and in the body of the manuscript (following instructions to authors). We also kept the full name (as I was always taught to do) when is in the beginning of a sentence. Missing italics have been checked.

Specific remarks:

  1. Introduction

- Page 2, lines 48-49: Species of the genus Phalacroma, producers of OA, should also be included.

Authors: Phalacroma species have been found to contain but not to produce OA. There is field evidence supporting the hypothesis that they are not producers de novo, buy act as vectors of the toxins contained in their ciliate prey (González-Gil et al. 2010), e.g., P.rotundatum fed the ciliate Tiarina fusus. We have added these explanations to the main body of the manuscript.

- Page 2, lines 78-79: DSP/LST intoxications are either non-fatal or rarely fatal, please revise.

Authors: “they are non-fatal syndromes”. The missing “non-” has been added

  1. Historical overview

- Page 4, lines 115-116: “… early shellfish poisoning outbreaks”. The authors should clarify what they mean by “early” outbreaks – does this refer to those in the very past (e.g. 60s)?

Authors: this has been changed to: shellfish poisoning outbreaks in the 70’s and 80’s

- Page 4, lines 117-119: As also mentioned later in the manuscript, Mesodinium is a ciliate and not considered as a HAB organism (or red tide one) – it is only a prey. This should be revised here by the authors.

Authors: We have clarified in the text that when we write “red tides” (a colloquially misused term to refer to any HAB event) we mean water discolorations (which are due to high concentrations of pigmented planktonic organisms. This is the correct meaning of this term. There are harmless red tides/water discolorations. The ciliate Mesodinium (M. rubrum, M. major) forms red tides (water discolorations) when very dense populations of this organism aggregate near the sea surface.

  1. Monitoring of lipophilic shellfish toxins in Chile

- There are multiple references to regulatory limits for LSTs (e.g. for OA, YTXs, etc.) throughout the manuscript, but the readers are not informed of what is actually regulated in Chile. The legislation/regulatory limits implemented in Chile should also be discussed at this point

- Page 5, lines 171-172: Completely obsolete EU legislation is referenced, please update with current legislation in force. The particular item cited as [35] has been repelled since 2004.

Authors: It was our mistake (Endnote typo) to add the link to the reference of the old 1991 EC directive. Now we have added references with the currently applied legislation in the EC, including the recent deregulation of PTXs (European Commission, 2021. In. Off J Eur Union L, 339:84-87) and the ongoing regulations in Chile, including:

  1. the EURLMB 2007. Standard Operating Procedure for detection of Okadaic acid, Dinophysistoxins and Pectenotoxins by Mouse Bioassay Version 4.0 (April 2007) European Union Reference Laboratory for Marine Biotoxins, based on
  2. Yasumoto et al 1984, Diarrhetic Shellfish Poisoning. In Ragelis E.P (ed.), Seafood Toxins. ACS Symposium series 262. American Chemical Society, Washington D.C: 207-214.

This is still the legal standard protocol used in Chile for determination of LST by MBA in seafood products.

- Page 5, line 209: “acquainted personnel”: knowledgeable or experienced personnel would be a better alternative.

Authors: “acquainted” have been changed to “experienced”

  1. Lipophilic Shellfish Toxin producers in Chile: distribution and toxic events 

- Page 8, line 316: “…DTX3 in shellfish above RLs (280 and 239 μg kg−1 HP”. No previous reference to existing RLs for Chile has been made - the authors should include this information earlier in the manuscript. Furthermore, “HP” (probably hepatopancreas” is an abbreviation not explained previously in the text. The authors should also clarify how this concentration in HP exceeds the RLs, since the latter are referring to shellfish meat (whole tissue) and not HP.

Authors: These data correspond to a research work using HPLC-FD (Alves et al. 2014), not to a monitoring analysis according to Chilean regulations. “R.L.” was deleted and the term HP explained. Changed to “Chromatographic analysis with fluorescent detector (HPLC-FD) of the mussels’ digestive glands (HP) in early summer (December) 2007 showed the presence of DTX1 and DTX3 (280 and 239 ng g−1 HP, respectively) associated…”

- Page 9, line 333: “…over 11,000 samples collected”: what kind of samples? it's probably shellfish, although not mentioned, as otherwise regulatory limits would not be applicable, nevertheless this should be clarified

Authors: Most of this para. have been re-written. There is a better description of the monitoring protocols applied in Chile for shellfish exports. These are intensively applied in Chiloé (Los Lagos), the island which is the main site of the Chilean mussel production.

- Page 9, line 334: “tested positive for LST”: tested positive commonly refers to exceeding regulatory limits. It would be better to use another expression, such as confirmed to contain LSTs or equivalent.

Authors: This expression has been changed to “confirmed to contain LSTs above detection levels (> LOD)

- Page 11, line 404: “majoritarian”: Better use another term, e.g. main, major, etc.

Authors: “majoritarian” has been changed to “main”

- Page 12, line 432: “They have been always assumed to be no relevant potential contributors”: please rephrase, this is not comprehensible.

Authors: changed toIt is assumed that these species do not contribute significantly to shellfish toxicity”.

- Page 12, line 444: “20,07”: do the authors mean “2007”?

Authors: Yes, our mistake. It has been corrected

- Page 15, line 502: Please provide a reference for the EU regulatory level for YTXs.

Authors: It has been added. But in the old MBA, YTXs are added to OAs and PTXs, and a single bioassay result (positive if mice die within 24 - 48 h after intraperitoneal injection of shellfish extract.

- Page 19, lines 633-634: “This microalga that has been found in the gastric content of browsers, e.g. limpets”: please rephrase, the syntax is confusing

Authors: Corrected to: “Empty cells of P. lima have been found in the gastric cavity of different species of browsing gastropods”.

  1. Lipophilic shellfish toxins and their accumulation and biotransformation by bivalve mollusks in Chile

- Page 19, line 664: “2.3 days)”: remove bracket. Authors: CORRECTED

- Page 20, lines 686-693: This section refers to toxins presence in bivalve molluscs. Information on toxin content of phytoplankton cells belongs elsewhere (in earlier sections of the manuscript).

Authors: Those data do not correspond to specimens from Southern Chile. They correspond to the same species (A. ostenfeldi) but from a nearby region in Argentina. It was necessary to put them here to support the view that accumulation of large amounts of spirolides in Chilean bivalves is not very likely (provided their strains were very similar to those from Argentina.

  1. LST new regulations and impacts on the Chilean shellfish industry, artisanal fisheries and other coastal commodities

- Page 21, line 750: “por”: “for” maybe? Authors: CORRECTED

- Page 22, lines 774-778: The authors should mention these specific directives. To my knowledge, EU Decision 2002/226/EC (to which this point probably refers) deals only with domoic acid content in pectenids. This practice is not yet regulated for LSTs.

Authors: Yes, we agree. But due to the ASP problem and the subsequent legal exception for scallops, these are now marketed after removing the digestive glands (that accumulates a large proportion ot the total toxin content). This benefits the toxin level results for any other toxin group. Now it reads:

“benefits from the exception of EC directives affecting control of amnesic shellfish poisoning (ASP) toxins in pectinid bivalves, i.e., only the adductor muscle and the gonad are to be analysed [++ ]. This control excludes the digestive gland, which is the organ that contains the highest proportion of lipophilic toxins, and consequently concentrations of any other kind of toxins in addition to ASP are greatly reduced”

- Page 22, line 782: “MB”: “MBA” maybe?

Authors: CORRECTED. Missing “A” has been added.

- Page 22, lines 795-796: “toxins planned to be deregulated soon (e.g.YTXs)”: Please provide a reference for this information. Is this within the EU, Codex, Chile?

Authors:  Nothing published about it, just rumours. We have deleted this comment.

  1. Conclusions and future perspectives
  2.  

- Page 22, line 807: “inambiguous”: “unambiguous” maybe?

Authors: CORRECTED

- Page 23, line 826: “lacking a phytoplankton”: “lacking a phytoplankton testing routine” maybe?

Authors: Corrected to “lacking a phytoplankton monitoring routine”